# IT Managers' Framing of IT Governance Roles and Responsibilities in Ibero-American Higher Education Institutions

Carlos Juiz [1], Francois Duhamel [2,*], Isis Gutiérrez-Martínez [3] and Luis Felipe Luna-Reyes [3,4]

1   Computer Science Department, University of the Balearic Islands, Carretera de Valldemossa, km. 7.5, 07122 Palma, Spain
2   Department of International Business, Universidad de las Américas Puebla, Ex Hacienda Sta. Catarina Mártir, Cholula, Puebla 72820, Mexico
3   Department of Business Administration, Universidad de las Américas Puebla, Ex Hacienda Sta. Catarina Mártir, Cholula, Puebla 72820, Mexico
4   Department of Public Administration and Policy, Rockefeller College of Public Affairs and Policy University at Albany 1400 Washington Av., Albany, NY 12222, USA
*   Correspondence: francois.duhamel@udlap.mx

**Abstract:** Present standards guiding the corporate governance of information technology (IT) provide useful frameworks for organizations' governing bodies to direct the effective use of information technology (IT) within their organizations. However, existing standards still fail to resolve the dilemma regarding the actual allocation of IT roles and responsibilities between governing bodies and IT management, while such an allocation represents a major challenge in many contemporary organizations. To advance on this issue, we explore IT managers' interpretation of the allocation of IT roles and responsibilities to either the governing body or managerial levels in nine Ibero-American Higher Education Institutions (HEIs). We used the ISO/IEC 38500 and COBIT standards to define a unique set of 212 management and governance activities and responsibilities. We surveyed 30 IT managers in Higher Education Institutions from nine Ibero-American countries and identified the divergence in the allocation of IT Governance and Management tasks between respondents and expert judgments. Using regression analysis, we show that the degree of such divergence depends on organizational contingency factors such as the formalization of IT procedures, centralization, the complexity of the organization, and the size of IT departments. This is the first study in the literature conducting a thorough analysis of IT task allocation between the governing level and the management level. This study is also the first to identify four organizational factors influencing the divergence between respondents and expert opinion regarding this allocation. The findings and propositions presented in this paper have the potential to extend our understanding of the IT governance dilemma in other professional organizations similar to HEIs.

**Keywords:** IT governance models; ISO/IEC 38500; COBIT; IT management; Higher Education Institutions; Ibero-American countries

## 1. Introduction

After more than 15 years of research in Information Technology (IT) governance, and the inception of the standard ISO/IEC 38500, there should not be doubt about the necessity of governing IT as a strategic asset [1–6]. IT governance is a set of mechanisms, i.e., structures, processes, and relational mechanisms, that support decision-making and alignment between IT and business [5,7,8] aimed at evaluating and directing IT activities and exercising control over them [9]. IT governance is concerned with promoting consistent and coherent decision-making behavior across the organization regarding IT in order to maximize the value that an organization derives from IT [1,10]. Without proper IT governance, contemporary organizations would be ill-equipped to respond to the challenges presented

by the increasing complexity of IT development and achieve business objectives [8,10–12], especially in Higher Education Institutions [6].

The ISO/IEC 38500 standard provides principles, definitions, and a model to govern IT in organizations [13]. The ISO/IEC 38500 standard is universally applicable, regardless of scale, ownership structure, and even the specific mechanisms, mentioned above, of the particular implementation. In particular, its IT governance model distinguishes between the governing body and the management level. Governing bodies comprise owners, directors, partners, or executives in an organization, responsible for the effective, efficient, and acceptable use of information technology within their organizations, directing, evaluating and monitoring the management level. The IT Managers label refers to the group of people responsible for the control and supervision of the use of IT assets and staff in their organizations or subunits, within the authority and accountability established by IT governance rules and delegated to them from governing bodies.

The governing body evaluates, directs, and monitors IT mechanisms for the organization whereas the IT management level plans, builds, and runs the IT-enabled business. Governing bodies and IT managers need to work together, accepting mutual accountabilities and responsibilities, to engage in efficient decision-making [11,14]. Governing bodies' implementation of best practices derived from IT governance standards is supposed to have a positive effect on the alignment between IT and the rest of the organization [6,10,12]. The way IT managers, who do not belong to governing bodies, understand their roles and responsibilities in their organizations is also crucial for the success of the implementation of whole IT governance frameworks based on standards.

However, the existing literature and standards do not provide sufficient guidance to practitioners yet, as the allocation of activities and responsibilities between the governing bodies and the IT management level allows for organizational inconsistencies in the implementation of the standard [15,16]. This is an important question to address as inconsistent concepts and implementation rules of IT governance [8] would leave participants with various dilemmas about the determination of "whom" is going to do "what", according to various contingency factors.

This type of allocation has not been extensively researched yet. Thus, it is timely and necessary to contribute to the literature and practice, developing an understanding of the way actors interpret the allocation of IT responsibilities between the two levels [4,5,17–19] upon the basis of a set of sound, well-accepted standards in this domain. Moreover, the present literature remains silent about the organizational factors influencing the allocation of activities and roles in IT governance, apart from a few exceptions such as [20] and [21], who do so but only limitedly so. Therefore, our research questions are as follows: What is the interpretation of the allocation of roles and responsibilities between the governing body and the management level in the implementation of IT governance frameworks from the perspective of IT managers? What are the factors influencing this interpretation?

Convergence in this domain, and the understanding of its determining factors, are preconditions to establishing efficient communication systems to connect both levels, and hence, to ensure effective implementation of IT governance frameworks faithful to the standard. Moreover, such an interpretation constitutes the main input for governing bodies (and other involved stakeholders) to assess the necessary capabilities to implement IT governance in their respective organizations.

First, we extracted 212 management and governance activities (best practices) from the existing IT governance standards. Due to the lack of a clear specification of "who does what" in these standards, we looked for an expert opinion to classify the 212 activities as managerial or governing body activities. We used this expert opinion as a reference point to measure the divergence between respondents and the expert, regarding the allocation of roles and responsibilities between the governing body and the management level. Thus, we measured the divergence between respondents and expert opinion in the perception of allocation of best practices between IT managers between governing and management

bodies. Second, we used standard OLS regression to estimate the impact of selected contingency factors on this divergence to address the second research question.

We address our research questions in this paper, taking the perspective of IT managers, in the context of Ibero-American universities, while most of the literature in IT governance takes the perspective of governance structures from higher hierarchical ranks [10,12]. We used data from two surveys conducted in 2019 and 2020 with Ibero-American Higher Education Institutions (HEIs) IT managers. Our results suggest that the allocation of roles and responsibilities between IT governance and IT management still represented a dilemma for Ibero-American IT managers at HEIs, in spite of ISO/IEC 38500 standardization efforts. Sources of divergence between respondents and expert judgments appear to be explained by organizational contingencies such as the formalization of IT procedures, centralization, the complexity of the organization, and the size of IT departments.

This research contributes to the literature by developing a practice-oriented vision of IT governance, focusing on the allocation of roles and responsibilities between the governing body and the management level in the implementation of IT governance, reflecting how IT managers in Ibero-American HEIs interpret this allocation. This study is also the first, as far as we know, to identify contingency factors prone to influencing the task allocation between those two levels. In spite of its importance, this allocation is not extensively discussed in the present literature. Thus, considering the issue of task allocation and its determinants represents a new direction in IT governance research with very concrete implications, as it directly addresses organizational challenges usually neglected in the present formulation of existing standards.

Although primary data are collected in the context of HEIs, we believe that findings and propositions introduced at the end of the paper may extend our understanding of the IT governance dilemma in other professional organizations similar to HEIs, such as hospitals and research centers, among others.

This article is structured in the following way. First, we start with a literature review dealing with essential functions in IT governance (governing body and management level), the integration of the relationship between governance and management functions in the standard ISO/IEC 38500, and particularly, IT governance at HEIs. Thereafter, the research methodology is explained. Then, the results of the empirical study are presented. In the discussion section, we confront our views with the present literature. Finally, in the conclusion, we indicate managerial implications, the limitations of the study, and future research directions.

## 2. Literature Review

In this literature review, we introduce the functions of governing bodies and those of IT management levels related to IT governance. We also discuss the difficulties of the implementation of IT alignment processes between the governing bodies and the IT management level, and the coordination and proper communication between those two levels. We finish the section with a brief note on the nature of IT governance choices in HEIs and the factors influencing them.

### 2.1. Functions of the Governing Body and Management Related to IT Governance

Many researchers have studied the implementation of mechanisms of IT governance and their impact on governing bodies–management relationships [9,14,21].

Governance activities ensure alignment between the functions, assets, and resources that they govern [11,20,22]. The main functions of the governing body are setting objectives and directing, evaluating, monitoring, and designing management incentives [23]. The governing body expects the implementation of the IT function from management. Hence, the governing body is expected to achieve the organization's purpose and objectives, as well as conformance to established norms of behavior [4,5]. Distinct governance structures may prevail [24], including, for example, a senior executive team [25].

IT management, on the other hand, operates in a context defined by the IT governance framework, under rules established by the governing body or delegates. Thus, IT management is accountable for the detailed aspects of tactical and operational planning, building and running the organization, and acting in the framework defined by the governing body [26].

Many articles have been devoted to the relationships between governing bodies and IT management during recent decades [18,19,27]. A large number of critical factors to ensure the success of IT governance has been identified, in relation to management support, internal management effects, structure, staffing, and the strategic alignment between IT and business [1]. Those factors, though necessary, may not be sufficient to ensure the success of IT governance in organizations [9,10]. Juiz and Toomey explained the difference between the process-oriented and behavior-oriented governance of IT, to understand why the governance of IT implementation may, in some cases, be overlapping with IT management processes [3].

The proper understanding of IT governance in an organization requires both a clear identification of the actors involved and a clear understanding of their roles and responsibilities [11,26].

### 2.2. Governing Bodies and IT Management in the Standard ISO/IEC 38500

The ISO/IEC 38500 model (Figure 1) rests on the distinction between governing bodies and management tasks: Governing bodies define the mission, policy, and strategy; appoint and oversee IT management; provide insight, wisdom, and judgement; and monitor performance. IT management, on the other hand, develops and delivers on the basis of policy strategy, sets and oversees operational business plans, appoints managers and staff, supports governance processes, implements board decisions, measures performance, and delivers services [25].

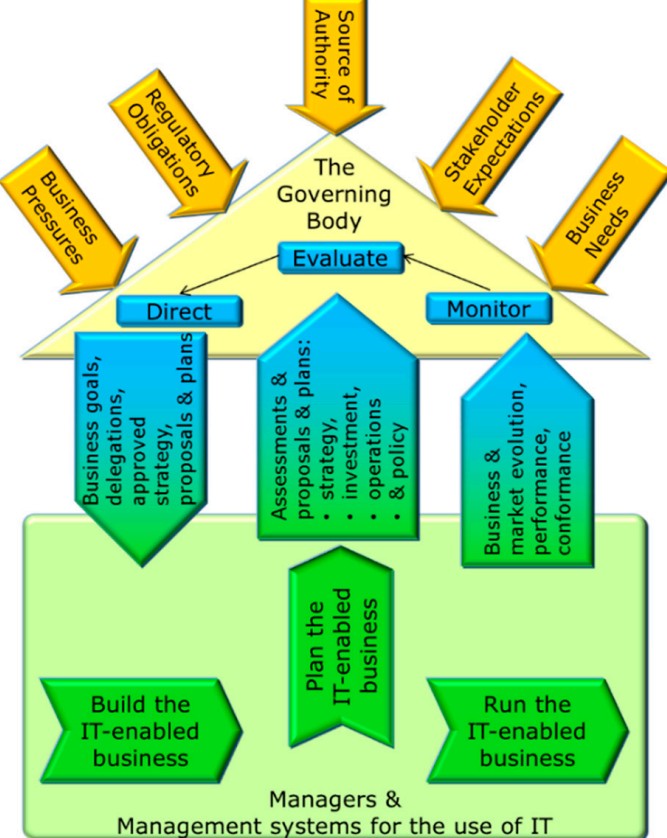

**Figure 1.** ISO/IEC 38500 model (own elaboration from Juiz and Toomey [3]).

Although the standard should provide clear guidance for establishing the relationships between governing bodies and management levels, ISO/IEC definitions usually obviate the issues out of its scope. Thus, the ISO/IEC 38500 model and standard, created from an IT governance perspective, intentionally omits IT management features present in the scope of other IT management standards (see Figure 2).

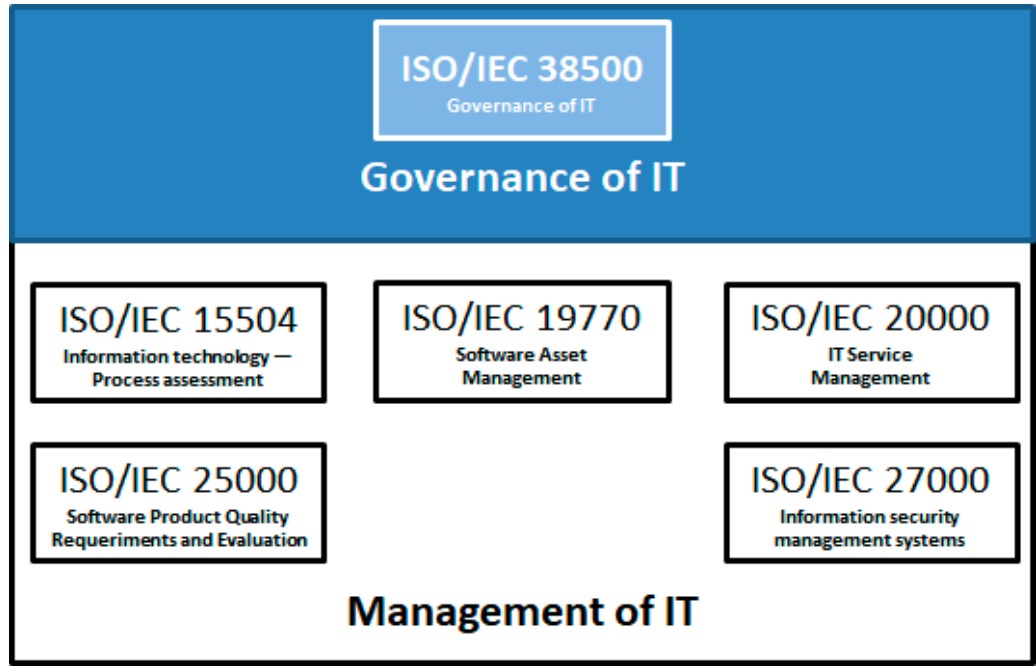

**Figure 2.** ISO/IEC IT Governance and main IT Management standard scopes (as appears in Juiz and Toomey [3]), ISO/IEC 38500 model (own elaboration from Juiz and Toomey [3]), ISO/IEC IT Governance and main IT Management standard scopes (own elaboration from Juiz and Toomey [3]).

In any case, The ISO/IEC 38500 establishes expectations for the relationship between governing bodies and management [28]:

- Responsibilities of the governing body: Members of the governing body are responsible for the governance of IT and are accountable for the effective, efficient, and acceptable use of IT within the organization.
- Responsibilities of IT managers: Managers are responsible for achieving organizational strategic objectives within the strategies and policies for use of IT set by the governing body.

In this general relationship, governance provides the means through which the governing body sets the strategic direction for the organization in respect of the use of IT. Jairak et al. identify the importance of joint committees comprising board members of both IT department and other parties. These members work together to establish the IT vision, mission, and strategies that are synchronized with the vision, mission, and strategies of the organization, to ensure IT/business strategy alignment [15].

Governance also monitors the state of the organization and the performance of its managers in achieving required outcomes. Effective governance of IT requires the establishment of an effective system of internal control as part of the organization's management systems. The standard also discusses issues of delegation [29].

However, as indicated before, ISO/IEC 38500 documents do not specify any design or implementation practices of an effective system for IT governance. Rather, the standard recognizes that different organizations can employ different implementation models, and that different models can be appropriate at different stages of the organization cycle [28]. This freedom in the implementation of the *de jure* standard of IT governance could be an enormous advantage compared to other more rigid proposals, but at the same time

could be its Achilles' heel, increasing uncertainty on the adequacy and alignment of local implementations when compared to the standard. That is, ISO/IEC 38500 falls short of providing guidance or best practices in making IT governance operational. The allocation of responsibility ("who is in charge") is often blurred by a propensity to inappropriately describe many IT management activities and processes as governance. Holt [29] considers that this is problematic as there is a myriad of ways through which the governing body and a management team can work together, combining decision-making models and the support of systems, processes, and procedures, in an integrated IT governance framework.

Therefore, understanding the perception of roles and responsibilities of both levels should be studied, especially given that a very strict set of rules for implementation of a standard is not necessarily the best solution.

### 2.3. Implementation of IT Governance Frameworks at HEIs

Higher Education Institutions (HEIs) have been working on the implementation of IT governance mechanisms and best practices even before the inception of the ISO/IEC 38500 standard [30]. However, the EDUCAUSE (ECAR) report highlighted that only 10% of the responding institutions had very effective IT governance programs and more than 60% had somewhat effective or even ineffective IT governance programs [31]. While IT governance seemed to be a critical issue in HEIs, few studies have been undertaken to examine the implementation of IT governance in this context and why its implementation could fail eventually.

Recent reviews of the literature include an overview of the majority of these studies [32,33]. IT governance among HEIs is diverse in its approaches. Some universities are using COBIT to implement an IT governance model, while other universities design their own models on the basis of the existing literature [34–37]. The model of the University of Calgary includes an architecture based on the creation of various committees, allocation of IT-related responsibilities and roles, risk management, and the use of a method for project management [38]. Fernández and Llorens designed and validated an IT governance reference model for universities, known within the Spanish university system as GTI4U, used in several Erasmus+ KA2 projects [39]. The coordinator of these EU projects has also created his own governance framework, based on the ISO/IEC 38500 model, known as dFogIT [16], implementing it in the University of the Balearic Islands over three years, with positive results [40]. The COBIT framework is based on an extensive list of processes, making it too cumbersome to implement [38], even more so in developing countries.

Regarding best practices, Jairak et al. identified a total of 65 practices as guidelines for the handling of IT governance issues under the principles of Sufficient Economy Policy (SEP) specific to the context of Thai universities. They show, more generally, the importance of the alignment between such practices and countries' socio-cultural values to support the implementation of IT Governance and IT standards in each context [15].

Organizations of any kind experience difficulties in implementing IT governance frameworks, producing a theory–practice gap in implementing IT governance in organizations [41–43]. The main reasons for such difficulties involve the lack of agreement on definitions of IT governance and its differences from pure IT management [8]. Moreover, there are barriers to implementing IT governance frameworks mainly due to the lack of communication between governing bodies and IT management structures, as well as different perceptions and measurements of IT values [44]. Another difficulty lies in the different interpretations given to the allocation of roles and responsibilities between the governing body and management level in implementing IT governance frameworks [29].

The gap between theory and practice in implementing IT governance may be determined by contingency factors pertaining to the organization and its environment. From a more theoretical point of view, classical structural contingency theory considers that the structure and results of an organization depend both on its own characteristics and on the environment in which it operates [45,46]. It indicates that task design and task allocation depend on a series of situational factors, that Mintzberg [23] identified as the age and

size of the organization, the technical system, the environment, and power factors, linked to the degree of centralization, complexity, and work formalization in the organization. In particular, the size of the IT department will influence the division and the degree of separation between the activities performed by governance and management structures. In smaller IT departments, with limited staffing and limited financial resources, responsibilities between governance and management tend to be blurred, so that governance structures have to get more involved in day-to-day management decisions [47]. Moreover, the level of formal knowledge of IT governance may influence the preference for organizations' strategic position with respect to IT.

Generally speaking, universities are considered professional bureaucracies [23], where behavior can be standardized by a coordinating mechanism, e.g., skills standardization, that allows for decentralization. In their operating core, Higher Education Institutions tend to grant autonomy to highly trained professionals in their work. Thus, operating units tend to be large. Managers typically share administrative tasks with the operating professionals. The administrative level is generally organized through steering/standing committees, task forces, and other liaison devices. The work of the operating professionals, because of its complexity, cannot be easily formalized, or its outputs standardized by action planning and performance control systems. This being said, there still exists a certain degree of variation regarding the organizational size, the degree of complexity, work formalization, and centralization in universities, accounting for different organizational outcomes between them.

## 3. Materials and Methods

This paper is based on the results of a survey applied to high-level IT managers from Ibero-American HEIs in 9 countries. The research was exploratory in nature, studying the perception of the allocation of roles and responsibilities of either governing bodies or IT management. Survey research is widely used to better understand individual perceptions of a given phenomenon [48].

### 3.1. Subjects and Sampling Approach

All survey participants in this study were involved in an international training program on IT governance practices. In December 2019, we started a quantitative exploratory study of IT governance involving 43 responses from IT managers (Group 1) from 43 Higher Education Institutions (HEIs) in 9 Ibero-American countries (Argentina, Brazil, Chile, Colombia, Ecuador, Mexico, Peru, Portugal, and Spain). Responses to the first round were aggregated by country. Then, to further refine the interpretation of the allocation of IT governance practices, we conducted another quantitative study in June 2020, recording individual responses from the 32 IT managers from 32 HEIs participating in the program (Group 2) in the same 9 Ibero-American countries. The questionnaire was distributed among the 32 participants, 30 of which responded with a fully valid questionnaire (93.8% response rate).

In Group 1, 100% of the subjects were IT managers. The size of the responding 43 HEIs, in terms of the number of students, ranged from 500 (a research center) to more than 356,000. The median size was 14,603 students. In Group 2, 100% of the subjects were also IT managers and the size of the responding 32 HEIs, in terms of the number of students, ranged from 100 to more than 109,000. The median was 11,603 students.

### 3.2. Data Collection

Following prior studies [49], a quantitative survey using questionnaires was chosen to be the main source of data. To design the questionnaires used in this study, the following considerations, described below, were implemented.

In the ISO/IEC 38500 model, the major difference between the governing body and the management level is that the governing body does not execute operations or perform management activities. Thus, the responsibility of the governing bodies is to be accountable

for their effective, efficient, and acceptable use within the organization, whereas managers' responsibilities consist in achieving organizational strategic objectives within the IT strategies and IT policies set by the governing body.

The interface between the governing bodies and IT management is identified in the ISO/IEC 38500 model through the connection between EDM (Evaluation, Direct, Monitoring) tasks and what management is doing (Plan–Do–Check–Act processes), but the connections through best practices are not explicitly formulated [28]. Therefore, the research questionnaire is based on the idea of confronting respondents (IT managers of Ibero-American HEIs) with a very large set of best practices. The respondents had to assign where the responsibility was allocated, either to the governing body or to the management.

In order to construct the questionnaire, we reviewed the standardization documents derived from ISO/IEC 38500, especially the IT governance assessment proposals in ISO/IEC 38501, 38502, 38503 (final draft standard), and 38504 [13]. This review was also extended to the actual cases of deployment of ITG4U [49] based on the enabling processes in COBIT 5 (updated to COBIT 19), the actual case of dFogIT implementation and dFogIT practices [16], and the assessment standard proposal for data centers known as ISO/IEC 22564 [50], which connects with some IT governance activities.

These source documents contained an extensive list of activities (best practices) that indicate "what" should be done (action) but sometimes without explicitly establishing the subject (actor) of the action (see Figure 3). Due to the intentional ambiguity of the ISO/IEC 38500 standard, it was necessary to mobilize an expert's interpretation of the allocation of activities between governing bodies and the IT management level, in the spirit of the standard, as a point of reference. This interpretation is based on a single expert, who is an internationally recognized expert in this area, acting as coeditor of the ISO/IEC 38503 standard, published in January 2022). This standard is precisely the current standard for assessing the governance of IT in organizations. The expert is also a member of the ISO committee for the current development of the ISO/IEC 30500 family of standards in Spain.

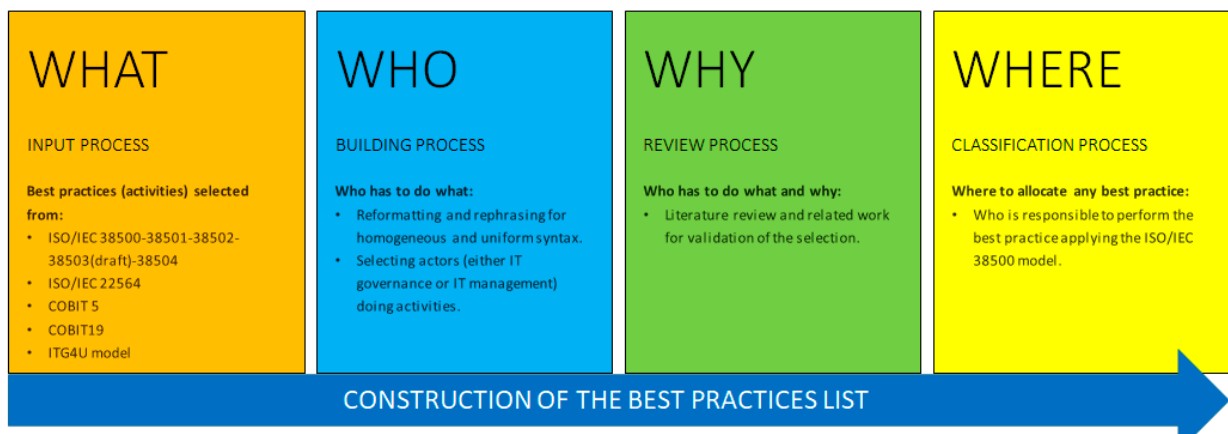

**Figure 3.** Model of classification of best practices allocation.

To construct the questionnaire, we applied a building process through which we explicitly assigned actors to activities, joining the "what" to do with the "who" is responsible, to form the "who has to do what" (see Figure 3). The process involved rewriting practices in a consistent, homologous, and coherent way. At the same time, we validated activities, through a review process, to ensure that each action constituted a best practice, in line with the key concepts previously explained in Sections 1 and 2. Therefore, all the actions that did not fit into the classification were discarded. Those selected as best practices were assigned to a structure level defined in the ISO/IEC 38500 model (where it should be allocated): Either to the governing body or to the management team. The classification process resulted in the formulation of 212 best practices, 111 corresponding to the responsibility of the governing body and 101 corresponding to managers. With the list of 212 best practices,

we clarify the responsibilities between governing bodies and IT management, not only indicating what activities should be performed as best practices, but also specifying who should be responsible for them (see Table A1 for the 212 practices).

In order to poll the subjects of Group 1 and Group 2, the resulting list of 212 best practices was anonymized again, eliminating who was responsible for what, and then submitted to survey participants, asking them to assign each practice as pertaining either to the governing body or to the management level. The questionnaire was applied after a short seminar on IT governance for IT managers, using the ISO/IEC 38500 model and standard. Participants in both groups were assigned to teams depending on their country of origin. Participants in Group 1 were asked to submit one response per team (Country), and participants in Group 2 were asked to submit both their individual response and an agreed response per team. We performed this questionnaire individually and then reached a consensus by countries with the two different samples (December 2019 and June 2020).

The questionnaire for Group 2 included 4 additional questions, asking participants to assess 4 organizational factors suggested to be important from the perspective of Contingency Theory: (1) The level of formalization of IT policy, (2) the complexity of the organization considering the number of organizational units and levels, (3) the degree of centralization of the decision making, and (4) the size of IT services relative to other divisions. Although we understand that using a single question to measure these factors has limitations, we decided not to add more questions to an already onerous 212-question survey. All 4 perceptions were captured with a single item with a 5-point Likert scale.

## 4. Results

This section of the paper includes a summary of key results from both Groups included in the study. We will briefly describe the results from Group 1 first, and then we will continue with the results from Group 2. Considering all 212 anonymized best practices, managers in Group 1 agreed with the expert 66% of the time. Managers in the group agreed with the expert in the classification of managerial and governance activities for 140.5 of the practices on average. On the other hand, the majority agreement among them was approximately 72%.

Table 1 shows the confusion matrix showing the main pattern of agreement and disagreement between the managers in the sample and expert judgement in more detail. There is more agreement than disagreement in the classification of practices. Respondents and experts agreed on the classification of 62.4% of the governance practices and 70.6% of the management practices. Our initial exploration suggested that the main areas of disagreement between the group and the expert were not random. Nonetheless, given that we did not have data for each individual in the sample, most patterns were inconclusive. In 37.6% of all practices, managers in the group misclassified managerial areas that the expert suggested to be governance activities, and managers assigned governance managerial tasks in 29.4% of the cases. We are calling these two types of divergence MaG divergence (managers appropriating governance activities) and MrM divergence (managers rejecting management activities).

**Table 1.** Confusion matrix with country teams' agreement with expert in Group 1.

| | | Country Team | | |
|---|---|---|---|---|
| | | **Governance** | **Management** | **Total** |
| **Expert** | **Governance** | 62.4% | 37.6% | 100% |
| | **Management** | 29.4% | 70.6% | 100% |

As described in the methods section, we asked members of the second group to classify practices into management and governance categories first individually, and then look for group agreement by the country team. Table 2 includes the confusion matrices showing the patterns of agreement and disagreement between respondents and the expert in Group 2. Table 2a shows the percentages using individual responses, and Table 2b shows the

percentage using country team responses. There are no significant differences between individual and group responses, and patterns are consistent with responses from Group 1.

**Table 2.** Confusion matrices with (a) individuals' and (b) country teams' agreement with expert in Group 2.

| | | (a) | | |
|---|---|---|---|---|
| | | **Individuals** | | **Total** |
| | | **Governance** | **Management** | |
| **Expert** | **Governance** | 64.3% | 35.7% | 100.0% |
| | **Management** | 29.6% | 70.4% | 100.0% |
| | | (b) | | |
| | | **Country team** | | **Total** |
| | | **Governance** | **Management** | |
| **Expert** | **Governance** | 68.0% | 32.0% | 100.0% |
| | **Management** | 26.8% | 73.2% | 100.0% |

Table 3 shows the list of the top 10 practices where managerial practices are assigned to governance by managers in the group (MrM divergence). The managerial practice most commonly assigned to governing bodies was "Someone formulates Human Behavior Policy and Plan," with 28 of the 33 managers responding to the questionnaire in this manner. In second place, there was a tie between "Someone ensures that the outputs of every level of organization and IT staff follow IT assets targets and drives the fulfillment of strategic objectives via improving the work performance of the organization and IT staff", "Someone implements a process for synchronizing business strategy and risk awareness of organization," and "Someone implements a process for the delegation of authority from governing body into management", with 23 survey respondents classifying them in that way. Other practices in this category include issues of strategic alignment, portfolio development, stakeholder satisfaction, and value creation.

**Table 3.** List of best practices with higher frequency of MrM divergence (classifying them as governance managerial activities).

| Item | Practice | F |
|---|---|---|
| 1 | Someone formulates Human Behavior Policy and Plan. | 28 |
| 2 | Someone ensures that the outputs of every level of organizations and IT staff are in accordance with the targets of IT assets drive realization of strategy targets via improving work performance of organizations and IT staff. | 23 |
| 3 | Someone implements a process for synchronizing business strategy and risk awareness of organization. | 23 |
| 4 | Someone implements a process for the delegation of authority from the governance body into management. | 23 |
| 5 | Someone analyzes satisfaction of stakeholders with IT projects and IT services. | 19 |
| 6 | Someone implements a process for strategic alignment with governance body directions. | 19 |
| 7 | Someone checks business work practices to ensure consistency with the use of IT. | 18 |
| 8 | Someone establishes an IT project, program and portfolio methodology for planning acquisitions. | 18 |
| 9 | Someone implements a process for evaluating, selecting and prioritizing IT projects. | 18 |
| 10 | Someone implements a process for making Health, Safety and Environmental (HSE) management strategies for physical environments, implement treatment measures, realize guarantee in terms of personnel, environments and etc., and avoid significant injury accidents of environments or personnel. | 18 |
| 11 | Someone implements a process to create new value by use of IT aligning the organizational strategy. | 18 |
| 12 | Someone keeps track of change management of strategic IT innovation. | 18 |

On the other hand, IT managers in the group also classified some governing body practices as managerial practices (see Table 4). Twenty-nine participants classified "Someone asks for an internal audit of IT services" as a management practice. Twenty-eight participants classified two other governing body practices as management practices, e.g., "Asking for a report of performance of IT regularly," and "Directing the design and publication

of procedures and regulations to implement IT policies." Other items in the list include issues of human resources management and training, planning in general and contingency planning in particular, implementation, and security assessments.

**Table 4.** List of best practices with higher frequency of MaG divergence (classifying them as management governance practices).

| Item | Practice | F |
|---|---|---|
| 1 | Someone asks for an internal audit of IT services. | 29 |
| 2 | Someone asks for a report of performance of IT regularly. | 28 |
| 3 | Someone directs the design and publication of a set of internal procedures and regulations that implement the previously defined IT policies. | 28 |
| 4 | Someone asks for a contingency plan for recovery IT services in the shortest time possible after a serious incident. | 27 |
| 5 | Someone evaluates whether enough human resources are available to undertake new IT initiatives, avoiding overloads. | 25 |
| 6 | Someone promotes training plan for IT usage. | 25 |
| 7 | Someone directs plans to be carried out according to the assigned IT responsibilities. | 24 |
| 8 | Someone ensures the effective implementation of each IT staff function and realization of management targets via set of organization structure and job responsibility. | 24 |
| 9 | Someone evaluates security reports and remediation of possible information leakage. | 24 |
| 10 | Someone evaluates security reports and remediation of not conformance with regulations. | 22 |

Table 5 shows descriptive statistics for the main variables included in the questionnaire to individual respondents. This table only includes results for all 30 IT managers that responded individually to all questions, including their perceptions of organizational characteristics in their universities. The first four variables in the table represent the participants' perceptions of the degrees of codification of IT procedures in their organizations, the complexity of the organization, the degree of centralization, and the size of the IT organizational unit. All four perceptions were captured with a single item with a 5-point Likert scale. Organizations in the sample tend to be on the lower side in terms of the codification of procedures and the size of the IT unit. They also tend to be perceived as complex with a degree of centralization on the higher end. The last three values in Table 5 include the average general agreement with the expert and the specific disagreements about the practice belonging to the managerial or governing body level. Participants agreed with the expert 68% of the time and classified managerial practices as governance 28% of the time. Moreover, participants in the group classified 36% of the governing body practices as managerial functions, according to the expert's perspective.

**Table 5.** Descriptive statistics of Group 2.

| Variable | Mean | Standard Deviation | Minimum | Maximum | n |
|---|---|---|---|---|---|
| Degree to which the work of the IT unit is codified into formalized procedures. | 2.83 | 0.99 | 1.00 | 5.00 | 30 |
| Complexity of the organization | 3.40 | 0.67 | 2.00 | 5.00 | 30 |
| Degree of Centralization | 3.70 | 0.65 | 2.00 | 5.00 | 30 |
| Size of IT Service/Department in comparison with other Services | 2.87 | 0.97 | 1.00 | 4.00 | 30 |
| Perc err MrM | 0.28 | 0.10 | 0.07 | 0.43 | 30 |
| Perc err MaG | 0.36 | 0.12 | 0.16 | 0.59 | 30 |
| Agreement | 0.68 | 0.08 | 0.55 | 0.82 | 30 |

Table 6 includes the correlation coefficients for all variables in the study. As expected, there is a negative and significant correlation of $-0.56$ between the general agreement between managers and the expert and the disagreements on managerial practices. Similarly, there is a negative and significant correlation of $-0.78$ between the general agreement and

disagreement on governing body practices. The only other significant correlation in the table is the one between the degree of task codification and size of the IT department, which can also be expected, given that larger IT areas are more likely to have the resources to codify their activities. All other correlations are not statistically significant.

**Table 6.** Correlations among variables.

| Variable | MrM Err | MaG Err | Agmnt | Cod | Cpx | Ctr | Srv |
|---|---|---|---|---|---|---|---|
| MrM err | 1 | | | | | | |
| MaG err | −0.09 | 1 | | | | | |
| Agreement (Agmnt) | −0.56 ** | −0.78 *** | 1 | | | | |
| Degree to which the work of the IT unit is codified (cod) | 0.27 | −0.13 | −0.06 | 1 | | | |
| Complexity of the organization (cpx) | 0.02 | 0.31 | −0.27 | −0.05 | 1 | | |
| Degree of Centralization (ctr) | −0.22 | 0.21 | −0.03 | −0.08 | −0.11 | 1 | |
| Size of IT Service/Department in comparison with other Services (srv) | −0.22 | −0.14 | 0.25 | 0.41 * | 0.24 | 0.04 | 1 |

* $p < 0.05$, ** $p < 0.01$, *** $p < 0.001$.

Finally, we ran exploratory OLS regressions, using the level of agreement and the two types of disagreements between IT managers and the expert as dependent variables (see Table 7). In terms of the general agreement, regression results suggest that agreement is negatively impacted by the complexity of the organization and positively impacted by the size of the IT department. In other words, it seems that distinguishing governing body's practices from managerial practices may become more difficult as the organization becomes more complex. On the other hand, larger IT departments may be more mature and therefore have more clarity in the distinction between management and governance.

**Table 7.** OLS regressions.

| Variable | Agreement | MrM Err | MaG Err |
|---|---|---|---|
| Degree to which the work of the IT unit is codified | −0.021 | 0.045 * | 0.001 |
| Complexity of the organization | −0.046 * | 0.019 | 0.070 * |
| Degree of Centralization | −0.014 | −0.023 | 0.048 |
| Size of IT Service/Department | 0.037 * | −0.044 * | −0.030 |
| Intercept | 0.843 | 0.301 | 0.026 |
| $R^2$ | 0.25 | 0.25 | 0.21 |
| N | 30 | 30 | 30 |

* $p < 0.05$.

The second model analyzes potential relationships between organizational variables and the MrM divergence, which involves assigning managerial practices to the governing body. As it is shown oinn Table 7, the degree of codification of practices within the organization appears to increase this type of divergence. The size of the IT department shows the potential to reduce the divergence. Finally, only the complexity of the organization has an impact on the MaG divergence, where managers assign to themselves what should be practiced pertaining to the governing body. We will discuss these results in detail in the next section.

## 5. Discussion

In the present discussion, we explore the sources of agreement and potential sources of divergence between respondents and expert judgments in task allocation, which constitute the dependent variable in our study. Then, for each of the four contingency factors we identified in the literature review and used for the quantitative study, we formulate, in this exploratory study, a series of propositions emerging from our exploratory statistical results.

*5.1. Managers' Perceptions of Task Allocation*

Potential areas of disagreement between participants and the expert opinion are listed in Tables 3 and 4. The following items include the top four practices with a higher frequency of MrM divergence (classifying managerial activities as governing bodies' activities).

- "Someone formulates Human Behavior Policy and Plan": General principles of Human resource policies are decreed by the governing bodies. However, the management level should be in charge of the detailed formulation (and implementation) of the plans inspired by those general principles. The goal is to ensure employees adopt the behavior expected by the governing body, in accordance with those general principles.
- "Someone ensures that the outputs of every level of the organization and the IT staff follow IT assets targets and drives the fulfillment of strategic objectives via improving the work performance of the organization and IT staff". IT staff's work performance is a management responsibility since the governing body is not directly responsible for IT staff/department/function. Once outputs are produced by management, the governing body may monitor and/or evaluate IT performance in general.
- "Someone implements a process for synchronizing business strategy and risk awareness of organization": Generally, processes are considered IT managing activities, all the more so when standards (such as the ones shown in Figure 2) are based on and developed for the sake of their implementation by managers. Some scholars consider that it should be the governing bodies' responsibility because this practice refers to the strategy, while others consider that it should be a managing issue, as it deals with implementation. This remains a controversial issue among scholars between process-based vs. principle-based governance [3].
- "Someone implements a process for the delegation of authority from the governing body into management": The same applies to this process and its implementation. Management is in charge of the delegation process and its implementation.

In those situations where discrepancies occur, managers seem to try to reject what sounds similar to strategic planning and/or policies that should belong to the governing body. However, the formulation, implementation, and assurance of those strategies' policies and plans are part of the managers' duties.

In Table 4, we selected the best practices with a higher frequency of MrG divergence (classifying governing bodies' activities as IT management activities).

- "Someone asks for an internal audit of IT services": IT managers may consider it as their own duty, but even though they execute (or subcontract) such an auditing process, the requirement should be coming from a superior layer in the organization, above the internal IT organization.
- "Someone asks for a report of performance of IT regularly": In the same way, IT performance (corresponding to the principles of the standard and COBIT) refers to organizational (or business) performance, not technical performance. IT performance should be monitored by the governing body (from IT management performance reports).
- "Someone directs the design and publication of a set of internal procedures and regulations that implement the previously defined IT policies": Direction of any activity regarding rules and norms is about governing, whereas planning and execution of these rules should belong to the managerial level.
- "Someone asks for a contingency plan for recovery IT services in the shortest time possible after a serious incident": The contingency plan for recovery IT services is clearly an IT management responsibility, but the governing body should solicit this plan from managers and give them some direction about the requirement of the shortest recovery of IT infrastructure and applications in order to the business continuity.

These top discrepancies in allocation may mean that IT managers try to overtake a power that is not claimed by governing bodies or executive teams, who should be asking for auditing, performance, continuity, etc., or for directing regulations on IT assets and

services to management. These questions remain in the realm of IT management because, usually, governing bodies are reluctant to direct, evaluate, or monitor IT [10].

In addition, both types of discrepancies can be explained by the lack of clear guidance about the relationship between IT governance and IT management in the standard documentation. The analysis from one ISO/IEC 38500 ad hoc group in 2020 also pointed out that the principles and governance model were not sufficient to provide unambiguous implementation guidelines.

In the following part, we formulate four propositions relating to the degree of agreement and divergence between IT managers and expert judgment for the whole sample to four organizational contingency factors, i.e., formalization, centralization, complexity, and size.

### 5.2. Formalization

The formal definitions of the nature of activities to be performed within an organization appears to lead to a higher degree of convergence between IT governance standards and managers' perceptions of the way tasks should be allocated between management and the governing body, thus reducing the latitude for the discrepancy between both and increasing organizational effectiveness [51]. However, divergence may also occur as a result of ambiguity in continuously evolving organizational norms. Codification may be incomplete or not attuned to particular needs, requiring flexible adaptation to unforeseen requirements stemming from personal situations or specific contexts, creating a gap between norms and actual practice, even with a high degree of codification [39]. The literature on communities of practice also shows that conventional bureaucracies are designed to solve stable problems for established constituencies through programs and policies and may not be sufficient in their structure to address complex, unforeseen occurrences [52]. Moreover, the strategy of IT governance deployment will vary depending on the flexibility of the academic regulation in each country. To apply the existing framework, each institution requires an adjustment period due to the rigidity of each framework [15]. If formalization is not adapted to practice, when people in charge are new in the organization, they may not even be aware of what they will have to do in practice, beyond the official prescription, requiring consultation with peers or with the executive level to make sure in case of doubt [53]. Our results tend to indicate that formalization may not represent a significant source of convergence to correct MaG-type divergence, but it does appear to be a source of divergence regarding MrM type divergence. Thus, we formulate proposition 1:

**Proposition 1.** *The more the degree of formalization of IT procedures in the organization, the more IT managers will attribute IT best practices pertaining to management to the governing body.*

### 5.3. Centralization

Organizations usually exhibit various degrees of centralization. When organizations are more centralized, more convergence is to be expected. In centralized organizations, there could be potentially less discretion from the management level to perform activities, increasing the propensity of managers to agree with standards regarding the allocation of tasks to the governing body. In this situation, centralization would act as a positive force to ensure an allocation of activities to the governing body consistent with the standard [53]. Meanwhile, a higher degree of centralization is usually associated with a lack of managers' empowerment [54]. It may encourage managers to turn away management tasks (according to the standard) to the governing body, as the management level would be less empowered to take responsibility over certain activities, based on prior experiences; management activities may also be "confiscated" by the governing body through "invasive" norms or tendency to authoritarianism.

The results of our study do not indicate any significant relation between centralization and agreement or divergence. Still, we consider the following proposition to be worthy of future investigation:

**Proposition 2.** *The more the degree of centralization of the organization, the more IT managers will attribute IT best practices pertaining to the management level to the governing body.*

*5.4. Complexity*

Complexity is attached to the number of hierarchical levels, entailing more protracted and bureaucratic decision-making processes. It may create cognitive stress, where organizational members, faced with overwhelming, complex decisions, show a series of behaviors stemming from ambiguity or confusion [55]. Thus, convergence between standard and practice would be more difficult to achieve in higher degrees of organizational complexity.

Increased organizational complexity may influence the relationship between governance and management in either direction. Complexity can encourage managers to try to reduce it by attributing to the governing body activities they should normally execute on their side. More hierarchical levels could increase the temptation to push management tasks to a superior level. On the other hand, managers may be tempted to widen their span of control to protect the areas under their responsibility, and control them, creating their own rules, as a strategy to cope with ambiguity [56]. Complexity would augment the likelihood of managers taking over activities normally pertaining to the governing body.

The results of the present study indicate that managers tend to appropriate for themselves more tasks belonging to the governing body than they should according to expert judgments. Hence, we formulate proposition 3:

**Proposition 3.** *The more the degree of complexity of the organization, the more IT managers will attribute to the IT management best practices pertaining to the governing body.*

*5.5. Size*

When IT areas are larger in size, or work with larger budgets, they tend to have more autonomy and may reclaim more activities for themselves [23]. Thus, they may be tempted to take over more activities usually pertaining to the governing body, especially when they benefit from more resources of different types and skills [57]. In this way, the divergence between standards and managers' perceptions of allocation regarding activities normally attributable to the governing body may be higher.

Meanwhile, when IT areas are larger in size, managers would have a better awareness of what their management responsibilities actually are, with fewer restrictions in terms of resources to implement them [57]. Larger firms could allocate additional financial and human resources to the implementation of an effective IT governance system, thus enhancing rigor and consistency [2]. In this way, a larger size should encourage better alignment between managers' perceptions and the standards regarding activity allocation as far as practices at the management level are concerned. Hence, we formulate proposition 4:

**Proposition 4.** *The bigger the size of the IT organization, the less IT managers will attribute best practices pertaining to IT management to the governing body.*

**6. Conclusions**

In this paper, we have shown that the allocation of roles and responsibilities to IT governance and IT management is still a dilemma for Ibero-American IT managers at HEIs, in spite of ISO/IEC 38500 standardization efforts. We found that there exists a significant divergence between IT managers in Ibero-American HEIs with the IT governance expert. In the second group, there were no significant differences between individual and country-level consensus. The patterns found were consistent with responses from the first group.

Thus, IT managers at HEIs tend to attribute to themselves more activities than they should actually do and sometimes do the contrary, which may suggest a lack of recognition of the value of IT governance, i.e., they do not fully acknowledge the idea of the separation of concerns pertaining to the IT governance concept, which is the core of the ISO/IEC 38500 model. However, IT governance arose two decades ago as IT management by itself was not

deemed sufficient to direct and control the growing sophistication and strategic integration of IT assets in the organization. The idea of separation into two levels in the ISO/IEC 38500 standards meets resistance from some organizations because of their organizational culture: Results appear less divergent in more mature HEIs where there have been already prior experiences of IT governance implementation in universities at the national level (e.g., in Spain, Portugal, and Brazil).

Even though the coincidence in the allocation of best practices with the expert was almost 70%, we accounted for two types of divergence: MaG divergence (managers appropriating governance activities) and MrM divergence (managers rejecting management activities). We explored the sources of agreement and divergence between respondents and expert judgments for each of the four variables we identified (formalization of IT procedures, centralization, the complexity of the organization, and size of IT function). From the statistical results we obtained, we also formulated, in this exploratory study, a series of propositions for further studies to analyze the divergences observed.

In responding to our research questions, we contribute to the literature by developing and documenting a practice-oriented vision of IT governance, reflecting how IT managers in Ibero-American HEIs interpret best-practices allocation between the governing body and the management level. To our knowledge, this study is the first in the literature to provide, on a quantitative basis, a description of how IT managers interpret the separation of responsibilities in which IT governance roles and responsibilities should be allocated, to either governing bodies or IT management levels, in universities. In addition, this study is the first to explore organizational factors influencing this allocation, formulating four propositions corresponding to each one of those dimensions.

In terms of practical implications, our results suggest that the ISO/IEC 38500 standard should entail a clearer prescription (in comparison to existing versions) of best-practices allocation between the governing bodies and the IT management level. IT managers' interpretation of this allocation stands as a precondition to efficient communication systems to connect both levels, and, hence, to be successful in any implementation of the IT governance model. Such an interpretation constitutes the main input for governing bodies and IT executive teams to assess the functionality and the degree of maturity of IT governance arrangements in their respective organizations. Our results also contribute to the proper identification and detection of potential problems in best-practices allocation at distinct organizational levels, according to organizational characteristics of HEIs and the environmental conditions they must address, as an essential basis for the implementation of IT governance frameworks, to avoid organizational conflicts and improve IT accountability. Thus, the configuration and deployment of IT governance mechanisms (structures, processes, and relationships) in organizations, particularly in HEIs, should specify who is in charge of given best practices, in terms of direction, planning, evaluation, checking, execution, and control. Our results could be used in other public organizations (e.g., government agencies or institutions), transferring the good practices suggested, with some adaptation to the specificities of the organization in question, according to variations in the contingency factors considered in this study.

In terms of research limitations, we only collected the perception of a limited sample of IT managers in HEI, concerning IT governance issues, and did not capture the perception from the higher hierarchy in HEIs on the same issues. However, members of the higher hierarchy are usually reluctant to assume the burdens of governing IT, which is the reason why IT governance standards exist in the first place.

Thus, in terms of future research directions, scholars should exploit the four proposals we formulated above on a larger sample and also survey higher hierarchical ranks from HEIs regarding their perceptions and behaviors of IT governance in their institutions.

**Author Contributions:** Conceptualization: C.J., F.D. and I.G.-M.; methodology: C.J., F.D., I.G.-M. and L.F.L.-R.; formal analysis: F.D. and L.F.L.-R.; data curation: L.F.L.-R.; writing/original draft preparation: C.J., F.D., I.G.-M. and L.F.L.-R.; writing/review and editing: C.J., F.D., I.G.-M. and L.F.L.-R. All authors have read and agreed to the published version of the manuscript.

**Funding:** The APC was funded by the University of the Balearic Islands.

**Institutional Review Board Statement:** Not applicable.

**Informed Consent Statement:** Not applicable.

**Acknowledgments:** We thank the MetaRed network for its support in the development of this research.

**Conflicts of Interest:** The authors declare no conflict of interest.

## Appendix A. List of 212 Practices Extracted from Current IT Governance Standards

**Table A1.** The questionnaire used in this survey is available upon request to the corresponding author of this article. We show, in the following table, the 212 practices examined in this research.

| Practice Description | Area of Responsibility |
|---|---|
| 1. Someone acquires IT assets, complying with standards and adapting to current and future use. | Management |
| 2. Someone aligns governance criteria for organization shaping the use of IT, regarding business strategy and reliance on IT, risk, compliance and decision-making model. | Governance |
| 3. Someone allocates responsibility, delegation of authority and accountability for IT-related decisions including principles, architecture, infrastructure and sourcing, solutions and investments. | Governance |
| 4. Someone analyzes satisfaction of stakeholders with IT projects and IT services. | Management |
| 5. Someone analyzes the satisfaction of stakeholders in relation to IT-based services in operations. | Management |
| 6. Someone analyzes to what extent IT contributes to the strategic goals of business units. | Governance |
| 7. Someone approves the organization's business strategy for IT. | Governance |
| 8. Someone asks for a Business Continuity Plan (BCP). | Governance |
| 9. Someone asks for a contingency plan for recovery IT services in the shortest time possible after a serious incident. | Governance |
| 10. Someone asks for a report of performance of IT regularly. | Governance |
| 11. Someone asks for an external audit of IT services. | Governance |
| 12. Someone asks for an internal audit of IT services. | Governance |
| 13. Someone asks for infrastructure and architecture plans to prevent IT obsolescence. | Governance |
| 14. Someone asks for IT acquisition planning. | Governance |
| 15. Someone asks for reporting about risks and security problems that may affect the continuity of services, so that they can decide on risk awareness and risk appetite for the organization. | Governance |
| 16. Someone asks for reporting of key performance indicators related to IT assets and strategy. | Governance |
| 17. Someone assigns responsibility for understanding the IT-related standards. | Governance |
| 18. Someone assigns the responsibility of being aware of IT-related legislation, norms and standards. | Governance |
| 19. Someone assigns the responsibility of directing and controlling IT assets to the CIO structure/office. | Governance |
| 20. Someone builds an IT governance framework considering IT and business market performance directions. | Governance |

**Table A1.** *Cont.*

| Practice Description | Area of Responsibility |
| --- | --- |
| 21.   Someone builds an IT governance framework considering stakeholders' interests. | Governance |
| 22.   Someone builds an updated reference catalogue that contains the IT-related standards applicable or already applied in the organization. | Management |
| 23.   Someone checks the emerging IT in the technological and business markets. | Management |
| 24.   Someone checks IT plans and policies to align with the organization's objectives in required timeframes and using allocated resources. | Management |
| 25.   Someone checks the level of IT skills of stakeholders. | Management |
| 26.   Someone checks business work practices to ensure consistency with the use of IT. | Management |
| 27.   Someone creates the architecture committee. | Governance |
| 28.   Someone creates the IT audit committee. | Governance |
| 29.   Someone creates outsourcing, out provisioning, etc., and other externalization policies committees. | Governance |
| 30.   Someone creates the risk policy committee. | Governance |
| 31.   Someone creates the structure (committee) for developing IT strategy and IT policy. | Governance |
| 32.   Someone defines how to continuously improve value of IT assets via new ideas and technologies. | Management |
| 33.   Someone defines and controls service and infrastructure components, maintains histories, plans and present statuses of service and infrastructure, keeps integrity and stability of IT assets. | Management |
| 34.   Someone defines and publishes a catalogue with all kinds of IT-related policies to guide the organization about IT implementation. | Governance |
| 35.   Someone delegates decisions about IT in a transparent and effective manner. | Governance |
| 36.   Someone designs a long-term program for implementing IT development. | Management |
| 37.   Someone designs a performance policy for business based on IT. | Governance |
| 38.   Someone designs a policy for IT projects and IT services benchmarking. | Management |
| 39.   Someone designs a professional career structure reflecting promotions based on the acquisition of IT skills and on successes obtained during change processes. | Management |
| 40.   Someone designs a set of IT policies aligned with the business strategy. | Governance |
| 41.   Someone designs a supplier relationship guide. | Management |
| 42.   Someone designs an acquisition policy. | Governance |
| 43.   Someone designs an IT governance framework considering laws and regulations. | Governance |
| 44.   Someone designs and disseminates a policy that promotes the general use of IT-related professional standards and best practices within the organization. | Management |
| 45.   Someone designs IT innovation policy. | Governance |

**Table A1.** *Cont.*

| Practice Description | Area of Responsibility |
| --- | --- |
| 46.  Someone determines if there is a need to review and when appropriate, revise the strategy for IT and associate policies. | Governance |
| 47.  Someone determines what information must be received to take decisions about IT performance. | Governance |
| 48.  Someone directs IT change organizational programs considering resources and skills, stakeholder involvement and responsibilities, budget and schedule, dependencies with business and prioritization of initiatives. | Governance |
| 49.  Someone directs plans to be carried out according to the assigned IT responsibilities. | Governance |
| 50.  Someone directs the design and publication of a set of internal procedures and regulations that implement the previously defined IT policies. | Governance |
| 51.  Someone ensures accumulation and inheritance of IT assets during the period of service lifecycle via creation, sharing and application of knowledge. | Management |
| 52.  Someone ensures enough resources to maintain quality and performance of IT services. | Management |
| 53.  Someone ensures its appraisal of external factors that may drive business opportunities and risk thereby mandating IT-related business change responses. | Governance |
| 54.  Someone ensures reasonable developments of IT assets by analyzing related parties' requirements making strategies that conform to the goals of IT resources, implementing and evaluating strategies as well as improving strategic capability of IT. | Management |
| 55.  Someone ensures that the availability of IT services meets demands of business operations and continues to optimize. | Management |
| 56.  Someone ensures that documents are in the condition of effective management by normalizing every activity during life cycle. | Management |
| 57.  Someone ensures that IT activities are consistent with identified Human Behaviors. | Management |
| 58.  Someone ensures that IT infrastructures and IT services can be restored within specific time after a disaster to support the overall business continuity requirements. | Management |
| 59.  Someone ensures that policies are developed to guide organizational behavior. | Governance |
| 60.  Someone ensures that service level agreements have been set up with all IT service users. | Management |
| 61.  Someone ensures that the organization has the IT-related capabilities required to support and sustain business operations. | Governance |
| 62.  Someone ensures that the organization's external and internal environments are regularly monitored and analyzed. | Governance |
| 63.  Someone ensures that the outputs of every level of organizations and IT staff are in accordance with the targets of IT assets, driving realization of strategy targets via improving work performance of organizations and IT staff. | Management |
| 64.  Someone ensures that there are mechanisms to clarify and interpret objectives, strategies and policies as emergent issues arise. | Governance |
| 65.  Someone ensures that there is a commitment and capability within the organization to undertake required changes. | Governance |

**Table A1.** *Cont.*

| Practice Description | Area of Responsibility |
| --- | --- |
| 66.    Someone ensures the effective implementation of each IT staff function and realization of management targets via set of organizational structure and job responsibility. | Governance |
| 67.    Someone ensures well-organized duty works as well as safe and stable operations of IT via standardizing responsibilities, working discipline and behaviors of duty work. | Management |
| 68.    Someone establishes a framework model for IT-related decisions, responsibilities and provision of information related to IT governance. | Governance |
| 69.    Someone establishes an IT governance framework considering board expectations. | Governance |
| 70.    Someone establishes an IT project, program and portfolio methodology for planning acquisitions. | Management |
| 71.    Someone establishes responsibilities for information structure and the intelligent analysis thereof from a strategic standpoint. | Governance |
| 72.    Someone evaluates appropriate costs for IT strategy. | Governance |
| 73.    Someone evaluates business satisfaction in relation to the use of IT. | Governance |
| 74.    Someone evaluates business strategy, business portfolios, risk awareness and business performance related to IT. | Governance |
| 75.    Someone evaluates gaps that require changes to achieve desired outcomes for the organization based on assessment criteria to evidence success/failure. | Governance |
| 76.    Someone evaluates integrity of information and protection of IT intellectual property. | Management |
| 77.    Someone evaluates IT capabilities and capacity management. | Management |
| 78.    Someone evaluates IT projects, programs and portfolios methodology. | Management |
| 79.    Someone evaluates IT services to realize approved proposals, balancing risks, and value for money of proposed investments. | Governance |
| 80.    Someone evaluates IT systems to ensure long-term business strategy. | Governance |
| 81.    Someone evaluates key aspects of organization related to IT assessments and decisions regarding business goals and strategy, risk appetite, performance, IT culture, IT maturity, training and competence, innovative use of IT, assurance reporting, key business processes IT supported and partner engagement. | Governance |
| 82.    Someone evaluates the options for providing IT. | Governance |
| 83.    Someone evaluates reports with the results of the internal and external audits, which clearly express the level of the organization's level of compliance with regulations and the risks that these entail. | Governance |
| 84.    Someone evaluates security reports and remediation of not conformance with regulations. | Governance |
| 85.    Someone evaluates security reports and remediation of possible information leakage. | Governance |
| 86.    Someone evaluates that IT supports achieving business objectives and risk appetite. | Governance |

**Table A1.** *Cont.*

| Practice Description | Area of Responsibility |
|---|---|
| 87. Someone evaluates that organizational use of IT complies with relevant laws, regulations. | Governance |
| 88. Someone evaluates that the business strategy makes the most effective use of IT to achieve business objectives. | Governance |
| 89. Someone evaluates value core of IT assets, create excellent cultural environments for sound developments, and provide powerful ideological and behavior guarantee by combing, implanting and continuously constructing organizational culture. | Governance |
| 90. Someone evaluates the consistency of Human Behavior in relation to IT activities. | Governance |
| 91. Someone evaluates the effectiveness of the IT Strategy in support of the Business Strategy. | Governance |
| 92. Someone evaluates the information that they need to meet their responsibilities and accountability. | Governance |
| 93. Someone evaluates the residual risk level within risk appetite of the organization. | Governance |
| 94. Someone evaluates the satisfaction of stakeholders with IT policies and strategy. | Governance |
| 95. Someone evaluates the segmentation of stakeholders for IT change processes. | Governance |
| 96. Someone evaluates whether enough human resources are available to undertake new IT initiatives, avoiding overloads. | Governance |
| 97. Someone evaluates whether IT governance processes are properly carried out in the organization. | Governance |
| 98. Someone evaluates whether IT projects and IT services take into account IT-related external regulations and laws and policies and internal procedures. | Governance |
| 99. Someone evaluates whether the organization conforms to its system (organizational policies and guidelines) for the Governance of IT. | Governance |
| 100. Someone formulates Human Behavior Policy and Plan. | Management |
| 101. Someone formulates the capacity planning strategy for IT assets. | Management |
| 102. Someone gathers business requirements and decides IT service level. | Management |
| 103. Someone implements a process for alignment between IT assets and IT capabilities. | Management |
| 104. Someone implements a process for assessing and evaluating risks of the current IT strategy. | Management |
| 105. Someone implements a process for assessing the risks associated with the use of IT during disaster recovery to address the continuing normal operations of business. | Management |
| 106. Someone implements a process for assigning accountability and delegation of competencies related to establishing the organization's performance indicators. | Management |
| 107. Someone implements a process for becoming aware of the IT-related needs and concerns of stakeholders. | Management |

**Table A1.** *Cont.*

| Practice Description | Area of Responsibility |
| --- | --- |
| 108.   Someone implements a process for building a Balanced Score Card for IT assets. | Management |
| 109.   Someone implements a process for building a catalogue of indicators to act on IT assets. | Management |
| 110.   Someone implements a process for carrying out project control in terms of scope, schedule, quality and cost based on the strategic targets of IT to ensure effective implementation of project and execution of strategic targets. | Management |
| 111.   Someone implements a process for checking competency of the assigned responsibility. | Management |
| 112.   Someone implements a process for checking effectiveness, efficiency, and acceptable use and delivery of IT in support of current and future business objectives. | Management |
| 113.   Someone implements a process for checking IT assets life cycle policies and processes. | Management |
| 114.   Someone implements a process for communicating IT-related internal policies and regulations to facilitate their dissemination in the organization. | Management |
| 115.   Someone implements a process for delegating decisions ensuring that the governance body is able to take final accountability. | Management |
| 116.   Someone implements a process for determining service catalogue and the agreed service level agreements with related parties, ensuring service capabilities meet requirements of related parties and are measurable. | Management |
| 117.   Someone implements a process for directing and communicating the need to meet the responsibilities and accountabilities. | Management |
| 118.   Someone implements a process for encouraging submission of proposals for innovative uses of IT. | Management |
| 119.   Someone implements a process for environmental reviews for preparing strategic plans for approval by the governance body including regulatory environment, technological advances, generational trends, skills availability, competitive forces, market development, stakeholder requirements and external threats. | Management |
| 120.   Someone implements a process for establishing review mechanism for significant incidents, controlling risks in advance, reducing operation risks of IT assets. | Management |
| 121.   Someone implements a process for evaluating, selecting and prioritizing IT projects. | Management |
| 122.   Someone implements a process for external audits to check whether IT projects and IT services comply with IT-related external laws and regulations and internal policies and procedures. | Management |
| 123.   Someone implements a process for formulating current and future business objectives related to use of IT (including IT infrastructure, IT services and IT delivery). | Management |
| 124.   Someone implements a process for identifying necessity of external laws and regulations as well as monitoring requirements for IT assets management, reasonably plan and implement to control potential risks. | Management |

**Table A1.** *Cont.*

| Practice Description | Area of Responsibility |
|---|---|
| 125.    Someone implements a process for identifying and analyzing risk factors arising from resistance to change or lack of commitment of stakeholders. | Management |
| 126.    Someone implements a process for implementing lifecycle management for architecture and technology, such as data, applications, and infrastructure, achieving balance between income and the risk introduced by the architecture and technology. | Management |
| 127.    Someone implements a process for improving fund application benefit and ROI (return on investment) via the management of budget and business accounting of IT assets in the case of financial compliance. | Management |
| 128.    Someone implements a process for including activities to mitigate risk related to a lack of commitment in IT projects. | Management |
| 129.    Someone implements a process for internal audits to check whether IT projects and IT services comply with IT-related external laws and regulations and internal policies and procedures. | Management |
| 130.    Someone implements a process for making Health, Safety and Environmental (HSE) management strategies for physical environments, implement treatment measures, realize guarantee in terms of personnel, environments and etc., and avoid significant injury accidents of environments or personnel. | Management |
| 131.    Someone implements a process for managing all kinds of change activities, controlling change risks, reducing impact of changes on production operation, and ensuring safety and stable operation of IT assets. | Management |
| 132.    Someone implements a process for managing risks in accordance with policies and procedures, escalated to relevant decision makers. | Management |
| 133.    Someone implements a process for measuring acknowledgement and understanding of IT policies. | Management |
| 134.    Someone implements a process for monitoring continuously IT projects and IT services in operation for cost control and financial performance. | Management |
| 135.    Someone implements a process for monitoring of disposal of assets and data. | Management |
| 136.    Someone implements a process for monitoring of IT budget and resource prioritization. | Management |
| 137.    Someone implements a process for normalizing IT human resource management of recruitment, training, appointment and retaining, ensuring staff meet the requirements of IT assets before, during and after appointment. | Management |
| 138.    Someone implements a process for normalizing supplier management, ensuring suppliers provide superior external technology resources and supports for IT assets. | Management |
| 139.    Someone implements a process for obtaining relevant information, properly sourced, collected, and analyzed to be presented to the governance body. | Management |
| 140.    Someone implements a process for realizing continuous improvement and promotion of service capability through the IT service identification of support business process and implementation of improvement. | Management |

**Table A1.** *Cont.*

| Practice Description | Area of Responsibility |
|---|---|
| 141.  Someone implements a process for reducing stakeholders' resistance to an IT-based change process. | Management |
| 142.  Someone implements a process for regular compliance assessment of IT use with relevant obligations, standards, and guidelines. | Management |
| 143.  Someone implements a process for restoring normal service operation within the shortest time, minimizing the negative impact of business operations, and ensure to keep service quality and availability level. | Management |
| 144.  Someone implements a process for selecting, evaluating and monitoring the IT acquisitions organization and suppliers. | Management |
| 145.  Someone implements a process for SLA establishment for suppliers and third parties. | Management |
| 146.  Someone implements a process for strategic alignment with governance body directions. | Management |
| 147.  Someone implements a process for synchronizing business strategy and risk awareness of organization. | Management |
| 148.  Someone implements a process for taking corresponding actions to improve effects of risk responses through measuring uncertainty and the influence on the targets. | Management |
| 149.  Someone implements a process for taking actions to eliminate deep causes to prevent recurrence of incidents or problems, reduce the impacts of repeatable incidents, and improve service quality and stability of IT assets. | Management |
| 150.  Someone implements a process for the delegation of authority from governance body to management. | Management |
| 151.  Someone implements a process for training related to the compliance of internal procedures with external laws and policies. | Management |
| 152.  Someone implements a process for training stakeholders in IT projects and services. | Management |
| 153.  Someone implements a process for updating IT governance information based on standards. | Management |
| 154.  Someone implements a process for updating IT management information based on standards. | Management |
| 155.  Someone implements a process of formulating the capacity planning strategy for IT assets. | Management |
| 156.  Someone implements a process to achieve real-time control of operation situation, and detect and solve abnormal operations via collection, classification and solving of application and operating information of IT infrastructures. | Management |
| 157.  Someone implements a process to create new value by use of IT aligning the organizational strategy. | Management |
| 158.  Someone identifies the roles and responsibilities related to IT governance and strategy. | Governance |
| 159.  Someone keeps track of change management of strategic IT innovation. | Management |

**Table A1.** *Cont.*

| Practice Description | Area of Responsibility |
| --- | --- |
| 160. Someone measures accurately IT spending. | Management |
| 161. Someone measures IT projects and IT services results. | Management |
| 162. Someone measures workload in IT projects and evaluates if appropriate. | Management |
| 163. Someone monitors alliances and collaborations with other organizations for data governance. | Governance |
| 164. Someone monitors appropriate and timely reporting on the evidence of success and change management. | Governance |
| 165. Someone monitors conformance reporting. | Governance |
| 166. Someone monitors for obtaining value from the use of IT. | Governance |
| 167. Someone monitors if there are deviations in service level agreements and corrective measures adopted. | Governance |
| 168. Someone monitors infrastructure and architecture obsolescence. | Governance |
| 169. Someone monitors IT investments plan and acquisition. | Governance |
| 170. Someone monitors IT projects current development and major drawbacks. | Governance |
| 171. Someone monitors level of uptake of IT management and IT governance standards. | Governance |
| 172. Someone monitors risk IT management reporting. | Governance |
| 173. Someone monitors that appropriate IT mechanisms for governance of IT are established. | Governance |
| 174. Someone monitors that IT risks identified related to Human Behavior are managed. | Governance |
| 175. Someone monitors that those given responsibility acknowledge and understand their responsibilities. | Governance |
| 176. Someone monitors the achievement of beneficial outcomes related to key aspects of IT deployment and use including business engagement, strategic alignment, business case realization, IT service delivery, service level and support, information security, risk, education and training. | Governance |
| 177. Someone monitors the level of knowledge concerning IT policies and laws in the organization. | Governance |
| 178. Someone monitors the performance of those given responsibility in the governance of IT. | Governance |
| 179. Someone monitors whether the inefficient use of IT affects its performance and communicates to stakeholders about how to correct it. | Governance |
| 180. Someone appoints special governance structures including Governance Steering Group, Risk Committee and Audit Committee. | Governance |
| 181. Someone plans acquisitions following directions from governance body. | Management |
| 182. Someone plans audit of IT assets to control potential risks of operation management. | Management |
| 183. Someone plans information security strategies and measures to reduce risk information assets face in the operation environments to acceptable level, so as to ensure availability, confidentiality and integrity of information. | Management |

**Table A1.** *Cont.*

| Practice Description | Area of Responsibility |
|---|---|
| 184. Someone promotes communication to disseminate the importance of IT governance. | Governance |
| 185. Someone promotes proper communication of IT policies. | Governance |
| 186. Someone promotes training plan for IT usage. | Governance |
| 187. Someone provides channels to receive user requests and standard services, provides users and customers with information and handling matters. | Management |
| 188. Someone provides leadership in developing strategies. | Governance |
| 189. Someone publishes a set of criteria for evaluating, selecting and prioritizing IT projects. | Governance |
| 190. Someone publishes an IT acquisition protocol including responsibilities for supplying information and decision-making. | Governance |
| 191. Someone publishes the benefits of IT projects and IT services. | Management |
| 192. Someone reduces or avoids deployment risks, decreases the number of incidents caused by the improper deploy of IT services. | Management |
| 193. Someone regularly analyzes the requirements of stakeholders. | Governance |
| 194. Someone regularly reviews which IT assets should be monitored by the board or should be delegated. | Governance |
| 195. Someone reinforces communication and relationship maintenance between IT staff and the related parties, such as customers, regulators or parent bodies, partners, suppliers, governments, etc., so as to realize mutual benefits. | Management |
| 196. Someone reports on IT Service Someone, Project Someone, Quality Someone, Resource management, supplier management process, IT Change Someone, IT Incident Someone and IT Cost Someone. | Management |
| 197. Someone reviews benefits and risks of externalization of services. | Governance |
| 198. Someone reviews security measures in place to maintain the integrity and quality of information. | Management |
| 199. Someone reviews stakeholders' participation in IT innovation. | Governance |
| 200. Someone reviews the acquisition policy, plans and relationships with suppliers and third parties. | Governance |
| 201. Someone reviews the financial resources to ensure IT innovation. | Governance |
| 202. Someone reviews the IT decisions, responsibilities and provision of information related to IT governance. | Governance |
| 203. Someone reviews the IT strategy plan. | Governance |
| 204. Someone reviews the long-term program of IT development. | Governance |
| 205. Someone reviews updated reference catalogue as compilation of IT-related regulations and laws that affect the organization. | Governance |
| 206. Someone runs the capacity planning strategy for IT assets. | Management |
| 207. Someone selects and prioritizes IT projects, programs and portfolios. | Governance |
| 208. Someone sets the responsibilities for evaluating emerging IT. | Governance |

**Table A1.** *Cont.*

| Practice Description | Area of Responsibility |
|---|---|
| 209.    Someone sets up a strategy structures (committees) to design the IT governance and strategy. | Governance |
| 210.    Someone takes into account any associated risk that might arise from strategy. | Governance |
| 211.    Someone takes into account the implications of the strategy for achieving business objectives. | Governance |
| 212.    Someone understands the business readiness for any major changes proposed as part of the business strategy. | Governance |

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
