# Peer review of "IT Managers’ Framing of IT Governance Roles and Responsibilities in Ibero-American Higher Education Institutions"

_informatics, doi:10.3390/informatics9030068_

Round 1

Reviewer 1 Report

Hi Authors,

Kindly find attached my thoughts.

Reviewer 2 Report

The paper has merits that can benefit the IT-governance practice. 

A few minor revisions are needed:

- The questionnaire used in the survey should be added as an appendix to the manuscript. This will be beneficial for practitioners to use the survey to evaluate their IT governance and IT management processes and practices. This also demonstrates how the 212 best practices are formulated in the survey. 

- The results (the ones shown in Tables 3 and 4) can be further extended to include all evaluations of the 212 best practices. The extended Tables can be added as an appendix. 

- The paper proposed 4 propositions from the research results. I would also suggest developing a proposition from the discussions in section 5.1. 

- The reasoning for reaching proposition 1 (section 5.2, p. 14) is very difficult to understand. If "formalization may not represent a significant source of convergence", then how IT managers will contribute more best practices as proposed in Proposition 1

- Proposition 2: The research results are based on the surveys of experts who are IT managers. How can the results support the perceptions of IT governance experts? Proposition 2 needs further clarification.

- Proposition 4: "the more the size of the IT organization" may change to "the bigger the size?

- The research contribution to IT governance research can be more elaborated and argued. How can academic researchers learn from the study? How do the study results advance the IT governance research, in which concrete ways? 

- Also how the research results can be used for revising the ISO standard? Since the standard is developed for corporate governance of IT, how this can be adapted to HEI? which are mostly public or state-owned HEI? And how the results can be used for other public organizations (e.g. government agencies or institutions?)

In summary, I think the paper has good results. A few minor revisions are necessary for improving the quality of the paper.
